# Factors Influencing International Infrastructure Investment: An Empirical Study from Chinese Investors

Senchang Hu ⬤, Yunhong Wang and Wenzhe Tang *⬤

State Key Laboratory of Hydroscience and Engineering, Institute of Project Management and Construction Technology, Tsinghua University, Beijing 100084, China; hsc21@mails.tsinghua.edu.cn (S.H.); yhwang_echo@163.com (Y.W.)
* Correspondence: twz@mail.tsinghua.edu.cn

**Abstract:** International economic cooperation accelerates the flow of capital, technology, labor, and other factors between different countries, which promotes global sustainable development. Building infrastructure construction is an important way to strengthen social development, and absorbing foreign capital is an effective way for developing countries to improve their infrastructure and to promote economic development. This study puts forward the factors that have influenced China's investment in international engineering projects, and it constructs a panel data regression model for empirical testing. The study shows that, first, international infrastructure investment tends to select countries or regions with good condition of highway infrastructure. Second, international infrastructure investment tends to choose countries or regions with low development level of port and power infrastructure. Third, bilateral diplomatic visits play a significant role in promoting international infrastructure investment. Fourth, international infrastructure investment tends to choose countries or regions with good resource endowment. This study reveals the influencing factors and the mechanisms for the choices of location for China's investment in international engineering projects, providing a theoretical framework for investors to optimize international infrastructure investment and management, as well as providing the policy references for developing countries to attract international infrastructure investment.

**Keywords:** international infrastructure investment; location choice; influencing factors

## 1. Introduction

Infrastructure construction is crucial to sustainable and coordinated economic, social, and environmental development [1–3]. On one hand, infrastructure is the foundation of social development, and specifically, strong infrastructure improves the quality of life as well as the health and well-being of residents by providing living, health, and education facilities such as roads, housing, electricity facilities, sanitation facilities, irrigation facilities, hospitals, and schools [4–6]. On the other hand, infrastructure is the motivation of economic development, for instance, infrastructure upgrades can reduce operating costs in economic activities, make it easier for people to access resources, and facilitate the learning of technologies [7,8]. High-quality infrastructure is an important prerequisite for the sustainable development of digital economy, smart cities, and Industry 4.0 technologies [9].

In developing countries or regions, the lack of reliable infrastructure has long been considered as a major factor that is hindering economic and social development [10]. For example, in sub-Saharan Africa, the lack of infrastructure has limited economic development and productivity improvement [11]. Many countries and organizations have recognized the importance of infrastructure and have prioritized infrastructure investment and construction [12,13]. When a country's public infrastructure cannot meet demand, and when the financial capacity of the government and the domestic private sector is limited, international capital is a viable option to employ to fill the infrastructure financing gap.

Considering that international investment is of great significance to infrastructure construction in developing countries or regions, China has actively fulfilled its responsibilities as a major country and has contributed its own strength to its infrastructure development. In "the Belt and Road Initiative" proposed by China in 2013, infrastructure connectivity was set as a priority [14–16]. The establishment of two major multilateral financial institutions under the Belt and Road Initiative, the Asian Infrastructure Investment Bank and the Silk Road Fund, aims to provide financial support and to enhance cross-border investment in countries. Investment and social capital from financial institutions can provide financial support for overseas infrastructure projects contracted by Chinese engineering enterprises, such as build-operate-transfer (BOT) projects, public–private partnership (PPP) projects, and finance–design–procurement-construction (F-EPC) projects [1]. A total of 65 Chinese Engineering enterprises entered the Engineering News-Record (ENR) Top 250 list of international contractors in 2017, as shown in Figure 1.

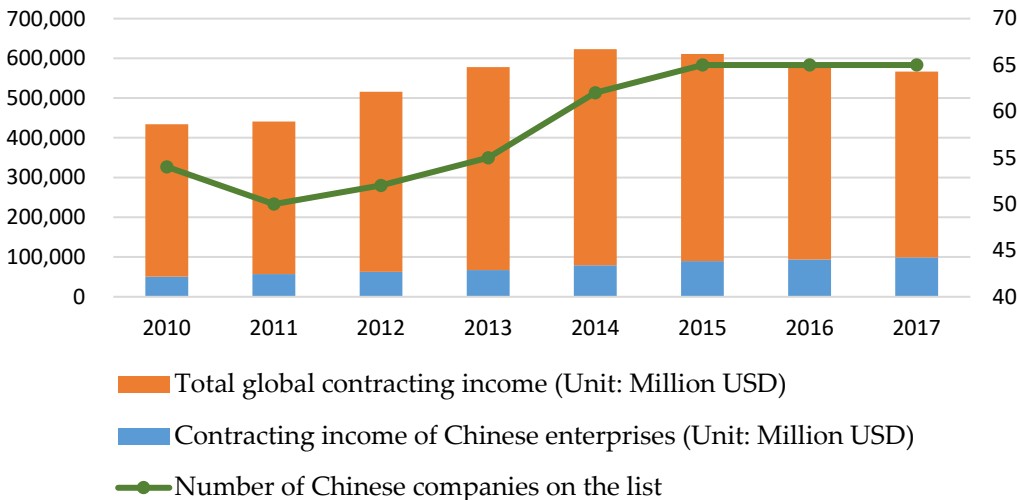

**Figure 1.** The ranking of Chinese engineering enterprises in ENR.

Chinese engineering enterprises' contracting income accounted for 21.1% of the global total income in that year, accounting for the highest proportion [17]. The contractor plays an important intermediary role in China's foreign infrastructure cooperation [18]. By analyzing the data and the experience of Chinese contractors and exploring the influencing factors of China's investment in international engineering projects, it will help improve the efficiency of China's foreign infrastructure investment.

Existing studies regarding China's investment in international engineering projects (CIIEP) lack considering the investment conditions of the host country [19,20] and mainly focus on individual project of a country [21,22]. This study identifies six major factors affecting international infrastructure investment, and accordingly, establishes a set of models on the factors influencing choice of international infrastructure investment location. With the support of data covering 134 countries and regions, the results of this research reveal the influences of infrastructure quality, bilateral diplomatic visits, and resource endowment on CIIEP. The outcomes of the study not only provide a theoretical framework for engineering enterprises to make decisions on international infrastructure investment but also have policy implications for developing countries to attract international infrastructure investment.

The rest of the study is structured as follows. Section 2 reviews the existing literature and presents the hypotheses regarding the factors' influences on international investment. Section 3 illustrates the empirical design. Section 4 addresses the findings and discussions. Section 5 summarizes the paper.

## 2. Literature Review and Theoretical Hypotheses

This section analyzes six factors, namely, diplomatic activities, resource endowment, commercial trade, infrastructure condition, institutional environment, and quality of finan-

cial markets, which can have an impact on international infrastructure investment, and summarizes the literature in the relevant areas. Based on the previous research results, six hypotheses are proposed separately.

### 2.1. Diplomatic Activities

Diplomatic activities are a barometer of bilateral relations, and they may have an impact on investment and cooperation between two countries [23]. Previous studies have focused on the role of diplomatic activities and political relations between the two countries. The quality of political relations between the home country and a host country is related to bilateral foreign direct investment [24]. According to the data of the United Nations Voting Conference and of China's import and export, Yi and Sun [25] predict that, although the improvement of bilateral political relations between China and other countries had a negative spatial spillover effect on commodity imports, China began to establish good bilateral political relations with potential importers.

As a matter of fact, China has attached importance to diplomatic activities and political relations between countries and has provided support for international engineering projects through diplomatic visits and contract signing. Diplomatic activities can directly bring opportunities for infrastructure investment and cooperation to Chinese engineering enterprises. In addition, diplomatic activities can help companies gain the support and protection of the host country to ensure the smooth implementation of the project. When enterprises carry out projects in host countries with high political and economic uncertainties, political endorsement and protection from the national level are particularly critical. Therefore, diplomatic activities may play a role in promoting CIIEP. We propose

**Hypothesis 1:** *Diplomatic activities can promote international infrastructure investment.*

### 2.2. Resource Endowment

Resource endowment is of great significance to economic growth because it can provide basic production factors for economic growth and it can help to cultivate industries related to resource endowment [26,27]. Wang et al. [1] analyzed the relationship between the energy resource endowment of the host country and China's foreign contracting projects, finding that the greater the per capita energy consumption of the host country, the greater the demand for power and chemical infrastructure construction, and thus the more projects China contracted for the host country. The essence of foreign infrastructure cooperation is the exchange of resources.

In the process of cross-border cooperation in infrastructure, a large number of foreign contracting projects can exchange more natural resources including crude oil, coal, natural gas, and metal ores. Correspondingly, China uses its advanced engineering technology and capability to exchange resources with the surplus natural resources of resource-oriented countries, and it helps some developing countries to carry out technological development and industrial development. In practice, Chinese companies tend to cooperate with countries with high resource endowments in their international engineering projects, such as oil-rich Saudi Arabia, coal-rich Mongolia, bauxite-rich Guinea, and iron-ore-rich South Africa. Therefore, resource endowment has also become an important factor in China's foreign economic cooperation. We propose

**Hypothesis 2:** *The resource endowment of the host country may promote international infrastructure investment.*

### 2.3. Commercial Trade

The business and trade of the host country may influence the location choice of international infrastructure investment. OFDI (outward foreign direct investment) could significantly improve exports [28]. Based on the data of 189 economies of the World Bank, Chen et al. [29] found that the regulatory factors and the business environment of the

host country would affect FDI (foreign direct investment) and that countries with stronger contract enforcement and more effective international trade regulations attracted more FDI. Wang et al. [1] found that the closer the bilateral trade relationship between the host country and China, the more cooperation occurred between China and the country in the field of infrastructure. Good commercial and trade activities and environment are the basis for economic cooperation between the two countries; trade development cannot be separated from necessary infrastructure construction [30].

The greater the demand for infrastructure in the active commercial trade activities, the greater the benefits that infrastructure investment can bring. China's investment in the construction of roads, ports, aviation, storage, and other infrastructure will facilitate commercial trade and will further promote the development of commercial trade. Many developing countries have a large import demand for manufactured products. Thus, China's cooperation with these countries in international engineering projects will help to build trust between the two countries and will set up a platform for commodity trade between the two countries, thereby promoting the export of China's manufactured products. We propose

**Hypothesis 3:** *Good business and trade in the host country may attract international infrastructure investment.*

### 2.4. Infrastructure Condition

The location choices of CIIEP may be influenced by the quality of local infrastructure. The measurement indicators of infrastructure conditions generally include the reliability and the efficiency of infrastructure services [31,32]. High-quality infrastructure can improve production efficiency [33,34]. High-quality infrastructure makes it easier to attract investment and business cooperation, reducing transaction costs for investors. For example, poorly constructed or maintained transportation systems may reduce the mobility of resources and the flexibility of economic systems [35], which can result in longer delivery times for goods and higher rates of damage to goods, and it can limit the use of more advanced transportation equipment [36]. Asiedu [33] argued that the reliability of infrastructure was more important than the availability of infrastructure for foreign investors.

On the other hand, infrastructure plays an important role in promoting the economic growth and industrial development of a country [37], especially for developing countries [38], since the backward infrastructure of these countries limits the potential of economic growth. From the perspective of market demand, countries with lower infrastructure conditions have a greater demand for infrastructure, and it is more urgent to develop their infrastructure systems and to improve their infrastructure conditions. Low-income countries lack sufficient budget to support infrastructure construction. Chinese enterprises can enter the infrastructure market of these countries with the support of foreign cooperation policies and can give play to their financial and technological advantages to help them improve the quality of infrastructure. It can be seen that countries with lower infrastructure conditions can provide Chinese enterprises with a larger infrastructure market. We propose

**Hypothesis 4:** *The relationship between the infrastructure condition of the host country and international infrastructure investment is uncertain.*

### 2.5. Institutional Environment

The location choice of international infrastructure investment may be influenced by the local institutional environment. Transnational infrastructure investment requires a stable, predictable, fair, and transparent institutional environment in the host country in order to ensure a stable return on investment [39,40]. Government policy uncertainty will have a negative impact on investment [33]. Infrastructure construction and investment projects have higher sunk costs [41] and are more sensitive to local policies. On the contrary, stable and trustworthy government policies will reduce the transaction costs of investment [42].

Effective government regulation, such as removing unfriendly market entry barriers [43] and controlling corruption [39,44], as well as improving the efficiency of approval, can create a favorable business environment for infrastructure investment enterprises.

Wisniewski and Pathan [45] found that foreign direct investors would avoid countries with excessive government expenditure, especially those with high military expenditure. Jiang and Martek [46] used the data of 74 developing countries from 2008 to 2017 to find that the political risk of the host country, such as investment risk, law and order risk, religious tension risk, and corruption risk, significantly affects foreign energy investment. In addition, protection of property rights and other legal institutional environments will have a positive impact on investment decisions, especially for infrastructure projects with a long payback period, such as hydropower stations and wind power plants [39,47]. The institutional environment may be one of the influencing factors for the location choice of CIIEP. We propose

**Hypothesis 5:** *The institutional environment of the host country will significantly affect international infrastructure investment.*

*2.6. Quality of Financial Markets*

Financial development is also an important part of the institutional environment. Existing studies have found that the financial development environment is an important factor for OFDI [48,49]. The location choice of CIIEP may be influenced by the local financial market environment. Bilir et al. [50] found that the financial development of the host country affects the investment of multinational corporations via both a financing effect and a competition effect. The content of the financial market environment is rich, encompassing aspects such as credit, capital flow, and exchange rate. The financial market environment of this paper refers to the acceptance of foreign ownership, the difficulty of equity financing, and the ease of obtaining loans.

A sound financial market environment can provide necessary financial support for investment, especially FDI. The developed financial market environment helps foreign-funded enterprises to obtain support from the government and from public and social groups, and it is easier to obtain funds through equity financing and loans. On the contrary, if the financial market environment of the host country is bad, the risks and obstacles for foreign enterprises to make investment will increase. Meanwhile, investment in international engineering projects is often characterized by large capital needs, high risks, and many uncertainties. The financial structure and the conditions of the host country can act as a stabilizer to help enterprises mitigate investment risks. Based on this, this paper further provides

**Hypothesis 6:** *Chinese enterprises may prefer to choose countries with favorable financial market environments when conducting international engineering projects.*

**3. Methodology and Empirical Design**

*3.1. Econometric Model*

In order to test the location selection factors of CIIEP, the following benchmark model was constructed, with reference to Yao et al. [51], to conduct panel data regression analysis on the empirical data.

$$y_{it} = \beta_{it} \cdot X_{it} + \gamma_{it} \cdot x_{it} + \delta_t + \varepsilon_{it}, i = 1, 2, \ldots\ldots, N, \quad t = 1, 2, \ldots\ldots, T. \tag{1}$$

where $y_{it}$ represents the explained variable; $X_{it}$ and $x_{it}$ represent the explanatory variables; $X_{it}$ represents the basic explanatory variables, including Visit, Treaty, Fuel, Ores, Manuf, and CoCV; $x_{it}$ represents the core explanatory variables—specifically, it includes infrastructure condition (QoRoad, QoPort, QoAir, QoElec), the institutional environment (PoS, GovE,

RegQ, RulL, ContrC), and the financial market environment (Owner, Equity, Loan); $\delta_{it}$ is the year fixed effect; and $\varepsilon_{it}$ is the error term.

In the process of empirical testing, variables are added step by step to ensure the robustness of the results. Model 1 to Model 5 on location selection and the influencing factors of CIIEP were constructed. Among them, Model 1 is the basic variable model, containing only the basic explanatory variables. Models 2–4 are the key variable models. On the basis of the basic variable model, the related variables of infrastructure conditions, institutional environment, and financial market environment are added. Model 5 is a full variable model, including basic explanatory variables and all key explanatory variables.

*3.2. Variable Description*

(1)   China's investment in international engineering projects (CIIEP). This study uses the data of China's foreign-contracted projects to measure CIIEP. Under the catalogue of foreign contracted projects, the statistics provide three indicators: the number of newly signed contracts, the fiscal amount of newly signed contracts, and the completed turnover. The newly signed contract amount and the completed turnover can reflect the scale of international engineering projects, so these two indicators are used for quantitative regression analysis [52,53].

(2)   Resource endowment and trade. In this paper, the indicators of the World Bank's Global Development Index (WDI) and the World Economic Forum's Global Competitiveness Index (GCI) are selected to measure the resource endowment and the business trade. Resource endowments are measured as a percent of fuel in merchandise exports and as a percent of ores and metals in merchandise exports [54,55]. Commercial trade is measured using the imports of manufactured goods and the business costs of policing crime and violence [56].

(3)   Diplomatic activities. This study selected bilateral diplomatic visits and bilateral contract signing to measure [57]. The bilateral diplomatic visit index counts the total number of visits by Chinese leaders to other countries and the number of visits by Chinese leaders to China each year; the bilateral contract signing index counts the total number of bilateral contracts signed between China and other countries each year.

(4)   Infrastructure conditions. The quality of infrastructure in this study is measured by the following four indicators: the quality of highway infrastructure, the quality of port infrastructure, the quality of aviation infrastructure, and the quality of power infrastructure [58,59]. These indicators cover the types of infrastructure that are most important to economic development.

(5)   Financial market environment. We use the prevalence of foreign ownership, financing through local equity market, and ease of access to loans to represent the financial environment [60].

(6)   Institutional environment. This study selects five Indicators from the Worldwide Governance Indicators (WGI) compiled by the World Bank to measure the institutional environment, including political stability, government effectiveness, regulatory quality, law rule, and corruption control [61,62].

(7)   Control variables. GDP growth rate, labor supply, and geographical distance are selected as control variables. Among them, the GDP growth rate and the labor force numbers are from the WDI Global Development Index, and the geographical distances are obtained from Centre détudes prospectives et d'informations internationals (CEPII) [63,64].

It should be noted that the number of newly signed contracts, the number of completed contracts, the size of the labor force, the import volume of industrial products, and the geographical distance are processed by adding 1 and taking logarithm to approach the normal distribution. The formula is expressed as follows:

$$V_{ij}^{*} = log_{10}(V_{ij} + 1). \tag{2}$$

where $V_{ij}$ represents the value of the $j$ indicator of the $i$ country. In addition, all data are normalized, and the formula is expressed as follows:

$$V_{ij}^{-} = \frac{V_{ij} - \mu_j}{\sigma_j}.$$

(3)

where $\mu_{\cdot j}$ represents the mean of the $j$ index, and $\sigma_{\cdot j}$ represents the standard deviation of the $j$ index.

### 3.3. Data

The panel data of this paper comes from empirical studies conducted in databases such as China Statistical Yearbook, China Diplomatic Yearbook, the World Bank, and the World Economic Forum [65]. Descriptive statistics and data sources of variables are shown in Table 1. The data cover nine years from 2009 to 2017, avoiding the impact of the financial crisis in 2008 and the trade conflict between China and the United States after 2018, and covering 134 countries or regions. All data come from the China Statistical Yearbook, the China Diplomatic Yearbook, and the Global Development Index (WDI).

**Table 1.** Descriptive statistics of variables and data sources.

| Variables | Abbreviations | Obs [a] | Min | Max | Mean | Data Source [b] |
|---|---|---|---|---|---|---|
| Value of Newly-signed Contract | VNC | 1023 | 0.00 | 248.53 | 11.15 | CSY |
| Value of Turnover Fulfilled | VTF | 1023 | 0.00 | 113.38 | 8.00 | CSY |
| Bilateral Diplomatic Visits | Visit | 1023 | 0.00 | 28.00 | 2.61 | CDY |
| Bilateral Treaty | Treaty | 1023 | 0.00 | 5.00 | 0.35 | CDY |
| Percent of Fuel in Merchandise Exports | Fuel | 1023 | 0.00 | 98.76 | 18.06 | WDI |
| Percent of Ores and Metals in Merchandise Exports | Ores | 1023 | 0.00 | 86.42 | 9.12 | WDI |
| Manufactures Imports | Manuf | 1023 | 0.11 | 1889.3 | 84.04 | WDI |
| Business Costs of Crime and Violence | CoCV | 1023 | 1.69 | 6.80 | 4.58 | GCI |
| Quality of Road Infrastructure | QoRoad | 1023 | 1.32 | 6.66 | 4.07 | GCI |
| Quality of Port Infrastructure | QoPort | 1023 | 1.27 | 6.81 | 4.23 | GCI |
| Quality of Air Transport Infrastructure | QoAir | 1023 | 1.06 | 6.87 | 4.58 | GCI |
| Quality of Electricity Infrastructure | QoElec | 1023 | 1.18 | 6.91 | 4.62 | GCI |
| Political Stability | PoS | 1023 | −2.81 | 1.59 | −0.05 | WGI |
| Government Effectiveness | GovE | 1023 | −1.54 | 2.27 | 0.22 | WGI |
| Regulatory Quality | RegQ | 1023 | −2.12 | 2.26 | 0.25 | WGI |
| Rule of Law | RulL | 1023 | −1.85 | 2.10 | 0.14 | WGI |
| Control of Corruption | ContrC | 1023 | −1.56 | 2.45 | 0.11 | WGI |
| Prevalence of Foreign Ownership | Owner | 1023 | 1.96 | 6.45 | 4.65 | GCI |
| Financing through Local Equity Market | Equity | 1023 | 1.10 | 5.94 | 3.58 | GCI |
| Ease of Access to Loans | Loan | 1023 | 1.25 | 5.74 | 3.10 | GCI |
| Growth of GDP | GDPG | 1023 | −20.60 | 25.16 | 3.08 | WDI |
| Quantity of Labor Supply | Labor | 1023 | 0.15 | 505.29 | 19.10 | WDI |
| Geographical Distance between Beijing and the Host Country | Distan | 1023 | 9.55 | 192.97 | 90.68 | CEPII |

Notes: (1) [a] The data cover years from 2009 to 2017. (2) [b] The specific sources of data are as follows. CSY: China Statistical Yearbooks, http://www.stats.gov.cn/english/Statisticaldata/AnnualData/ (accessed on 6 January 2022); WDI: World Development Indicators, https://datacatalog.worldbank.org/dataset/world-development-indicators/ (accessed on 6 January 2022); GCI: Global Competitiveness Index, http://widgets.weforum.org/global-competitiveness-report-2017/ (accessed on 6 January 2022); WGI: Worldwide Governance Indicators, http://info.worldbank.org/governance/wgi/ (accessed on 6 January 2022); CDY: China Diplomatic Yearbook, https://navi.cnki.net/knavi/yearbooks/YZGWJ/detail (accessed on 6 January 2022); CEPII: Le Centre d'études prospectives et d'informations internationals, http://www.cepii.fr/ (accessed on 6 January 2022).

At the same time, this study adopts the balanced panel data regression method for quantitative analysis. The selected panel data is obtained by repeated observation of the same cross-section, which contains information of the individual dimension and the time dimension. It is suitable to use the fixed-effect panel data regression method for analysis and to fix the correlation of individual errors over time [65].

On this basis, the new contract number of China's foreign contracting projects (VNC) is used as the explained variable for regression analysis, and the completed contract amount (VTF) is used as the explained variable for the robustness test of the model. We conducted unit root tests on the explained variables VNC and VTF to verify the stationarity of the data.

## 4. Empirical Results and Discussions

### 4.1. Data Stationarity Test

Before regression analysis, it is necessary to verify the stationarity of the data to ensure that there is no problem of spurious regression [66]. This is shown in Table 2. The result of unit root test shows that the explained variable is significant at the level of 0.001, indicating that the panel data in this paper has stationarity.

**Table 2.** Unit root test results.

| Variables | Coefficient | *p*-Value | Stationarity |
|---|---|---|---|
| VNC | −7.8598 *** | <0.001 | stable |
| VTF | −7.7363 *** | <0.001 | stable |

Notes: *** $p < 0.01$.

### 4.2. Regression Analysis Results

First, we compared the results of Models 1–5 in Table 3 for Visit, Treaty, Fuel, Ores, Manuf, and CoCV. Table 3 provides the estimation results of the fixed effects model.

(1)  Diplomatic activities. In Models 1–5, the coefficients for Visit were all significantly positive; those for Treaty were positive, but not statistically significant. The results listed above show that diplomatic activities do affect CIIEP, and that this influence is mainly manifested through bilateral diplomatic visits. The more active the bilateral diplomatic visits are, the closer China's cooperation in the field of infrastructure will be. However, the signing of bilateral contracts has no significant impact on CIIEP. This result provides some evidence for Hypothesis 1. Compared to previous studies [67], this result shows that bilateral diplomatic visits can promote the rapid development of economic cooperation between countries in the short term, but it takes some time for the signing of contracts to be implemented, which fails to produce a significant promoting effect.

(2)  Resource endowment. In Models 1–5, the coefficients of Fuel and Ores were significantly positive, indicating that CIIEP significantly tends to choose resource-rich countries or regions, especially fuel resources, ores, and metal resources, which verifies Hypothesis 2. Regions with good resource endowment can often provide necessary resources for infrastructure construction, which is conducive both to attracting foreign capital inflow and to promoting infrastructure development [68,69].

(3)  Commercial trade. In Models 1–5, the coefficient of Manuf was unstable, indicating that there is no significant relationship between CIIEP and the import volume of industrial goods in the host country. The same as some previous research findings [70], the coefficient of CoCV was significantly negative, indicating that CIIEP tends to choose countries or regions with lower costs for public security prevention and commerce; this verifies Hypothesis 3.

Meanwhile, the infrastructure condition in Model 2 and in Model 5 in Table 3 was compared, including the calculation results of QoRoad (highway infrastructure condition), QoPort (port infrastructure condition), QoAir (aviation infrastructure condition), and QoElec (power infrastructure condition). The regression coefficients of QoRoad, QoPort, and QoElec were significant in Model 2 and in Model 5, while the regression coefficients of QoAir were not significant in Model 2, indicating that the infrastructure condition of the host country will significantly affect CIIEP, which verifies Hypothesis 4. First, the regression coefficients of QoRoad were all significantly positive, indicating that CIIEP significantly tends to choose countries or regions with good highway infrastructure conditions. Secondly,

the regression coefficients of QoPort were all significantly negative, indicating that CIIEP significantly tends to choose countries or regions with low port infrastructure conditions. In addition, the regression coefficients of QoElec were all significantly negative, indicating that CIIEP significantly tends to choose countries with a low quality of power infrastructure.

**Table 3.** The regression results of model 1–5.

| | VNC | | | | |
|---|---|---|---|---|---|
| | **Model 1** | **Model 2** | **Model 3** | **Model 4** | **Model 5** |
| Visit | 0.088 ** | 0.089 ** | 0.080 ** | 0.091 ** | 0.083 ** |
| | (3.01) | (3.08) | (2.79) | (3.16) | (2.98) |
| Treaty | 0.034 | 0.041 | 0.021 | 0.040 | 0.032 |
| | (1.35) | (1.64) | (0.84) | (1.58) | (1.32) |
| Fuel | 0.236 *** | 0.248 *** | 0.210 *** | 0.249 *** | 0.207 *** |
| | (9.99) | (10.48) | (8.29) | (9.97) | (8.17) |
| Ores | 0.113 *** | 0.118 *** | 0.128 *** | 0.108 *** | 0.109 *** |
| | (4.59) | (4.81) | (5.26) | (4.43) | (4.53) |
| Manuf | −0.106 ** | −0.085 | −0.022 | −0.193 *** | 0.024 |
| | (−2.83) | (−1.49) | (−0.36) | (−4.33) | (0.38) |
| CoCV | −0.051· | −0.066 * | −0.078 * | −0.073 * | −0.060· |
| | (−1.68) | (−2.05) | (−2.31) | (−2.43) | (−1.74) |
| *GDPG* | 0.130 *** | 0.125 *** | 0.120 *** | 0.105 *** | 0.063 * |
| | (4.94) | (4.78) | (4.66) | (3.91) | (2.38) |
| *Labor* | 0.553 *** | 0.522 *** | 0.495 *** | 0.562 *** | 0.407 *** |
| | (14.54) | (11.23) | (9.51) | (13.62) | (7.49) |
| *Distan* | −0.047· | −0.041 | −0.067 * | −0.053· | −0.081 ** |
| | (−1.74) | (−1.48) | (−2.37) | (−1.87) | (−2.84) |
| QoRoad | | 0.197 *** | | | 0.116 ** |
| | | (4.46) | | | (2.64) |
| QoPort | | −0.093 * | | | −0.126 ** |
| | | (−1.97) | | | (−2.78) |
| QoElec | | −0.163 *** | | | −0.100 * |
| | | (−3.48) | | | (−2.16) |
| QoAir | | 0.073 | | | 0.092· |
| | | (1.47) | | | (1.79) |
| PoS | | | 0.045 | | 0.063 |
| | | | (1.12) | | (1.55) |
| GovE | | | 0.289 ** | | 0.165 |
| | | | (2.71) | | (1.45) |
| RegQ | | | −0.583 *** | | −0.610 *** |
| | | | (−7.92) | | (−7.76) |
| RulL | | | 0.079 | | −0.030 |
| | | | (0.64) | | (−0.23) |
| ContrC | | | 0.111 | | 0.152· |
| | | | (1.21) | | (1.68) |
| Owner | | | | 0.010 | 0.106** |
| | | | | (0.23) | (3.03) |
| Equity | | | | 0.153 *** | 0.139 *** |
| | | | | (4.19) | (3.59) |
| Loan | | | | −0.021 | 0.021 |
| | | | | (−0.51) | (0.51) |
| Year FE | Yes | Yes | Yes | Yes | Yes |
| N | 1023 | 1023 | 1023 | 1023 | 1023 |
| R Square | 0.438 | 0.456 | 0.477 | 0.452 | 0.510 |

Notes: *** *p* < 0.01, ** *p* < 0.05, * *p* < 0.1; Year FE = Year fixed effect; N represents the number of observations

Subsequently, the institutional environments of Model 3 and Model 5 in Table 3 were compared, specifically including the results of PoS (political stability), GovE (government effectiveness), RegQ (regulatory quality), RulL (legal perfection), and ContrC (corruption

governance). On the whole, the regression coefficients of the PoS, RulL, and ContrC variables in Model 3 and Model 5 were not significant, the regression coefficients of GovE in Model 3 were significantly positive, and the regression coefficients of RegQ in Model 3 and Model 5 were significantly negative. These results indicate that China's investment in international engineering projects is significantly inclined to choose countries and regions with poor quality of government supervision. The poor quality of government supervision means that various constraints, such as environmental regulations and laws and regulations, are lessened, and infrastructure construction and related economic cooperation can be carried out more conveniently.

Finally, we compared the financial market environment of Model 4 and Model 5 in Table 3, specifically including the results of Owner (acceptance of foreign ownership), Equity (local equity financing), and Loan (ease of access to loans). The coefficients of Owner and Equity were significantly positive, while the regression coefficient of Loan was not significant, indicating that CIIEP significantly tends to choose countries and regions where Equity financing is easier, which verifies Hypothesis 6. International cooperation in infrastructure needs a large amount of financial support. Low difficulty in equity financing will help China's overseas infrastructure investment to obtain necessary financial support and will promote international economic cooperation.

*4.3. Discussion*

1.  The impact of the infrastructure condition

    The analysis results of models show that the quality of a host country's infrastructure has a significant impact on CIIEP.

    First, CIIEP tends to choose countries or regions with good highway infrastructure conditions. In other words, countries with poor quality road infrastructure face greater difficulties in attracting infrastructure investment, which indicates that high-quality road infrastructure is very basic and necessary for infrastructure construction activities, including the transportation of machinery and materials needed for construction activities. Infrastructure construction in the environment of poor-quality highway infrastructure will produce more project cost and more time limit pressure.

    Hence, for low-income developing countries facing the pressure of public expenditure, the government should consider highway infrastructure construction to be the key development priority and then invest the limited budget in road infrastructure construction first, so as to lay a good foundation for attracting more infrastructure investment in the future. Developmental financial institutions such as the World Bank and the Asian Development Bank should pay attention to the importance and the particularity of road infrastructure, should invest in road infrastructure in low-income developing countries, and should shoulder development responsibilities.

    Second, there is a significant negative correlation between port infrastructure conditions and CIIEP. CIIEP significantly tends to choose countries or regions with low port infrastructure conditions. Maritime transportation is the long-distance trade transportation mode with the lowest transportation cost. Chinese enterprises continue to investigate and track countries or regions with low port infrastructure conditions and focus on port construction or investment opportunities.

    Finally, there is a significant negative correlation between the quality of power infrastructure and CIIEP. CIIEP significantly tends to choose countries or regions with a low quality of power infrastructure. Sufficient and stable power supply is of great significance to national living standards and to national industrial development. For example, the Bishkek Thermal Power Plant renovation project in Kyrgyzstan, the Jimpur Wind power project in Pakistan, the Karot hydropower project in Pakistan, and so on. These projects are helping countries with low-quality power infrastructure gain momentum.

2.  The influence of diplomatic activities

Bilateral diplomatic visits have a significant positive impact on CIIEP. The more active the bilateral diplomatic visits are, the higher the new contract amount of China's foreign contracted projects in the country will be. The results reveal that high-level diplomatic visits can build platforms, create opportunities, and sign contracts to directly promote cooperation on infrastructure projects.

3. The impact of resource endowment

CIIEP obviously tends to choose resource-rich countries or regions, including those with fossil fuel resources, ore resources, and metal resources, which confirms that resource exchange is an important motivation of CIIEP. Building long-term partnerships with countries rich in natural resources through infrastructure projects could help China gain more resources for its future development. Resource-based countries can make use of their own resource advantages and can strengthen cooperation with China in the field of infrastructure, so as to transform their resource advantages into infrastructure advantages.

*4.4. Robustness Test*

In order to verify the robustness of the above research results, this study adopted the method of transforming the explained variables. We replaced the original number of new contracts signed (VNC) with the number of completed contracts (VTF), kept the remaining variables unchanged, and re-conducted the fixed-effects panel data regression analysis; the results are shown in Table 4 below.

**Table 4.** Robustness test results.

|  | Model 1 (VTF) | Model 5 (VTF) |
|---|---|---|
| Visit | 0.083 ** (2.89) | 0.075 ** (2.77) |
| Treaty | 0.034 (1.36) | 0.036 (1.55) |
| Fuel | 0.261 *** (11.28) | 0.232 *** (9.39) |
| Ores | 0.133 *** (5.51) | 0.131 *** (5.59)) |
| Manuf | −0.077 * (−2.08) | 0.004 (0.07) |
| CoCV | −0.059 * (−2.01) | −0.082 * (−2.45) |
| GDPG | 0.134 *** (5.16) | 0.068 ** (2.66) |
| Labor | 0.532 *** (14.25) | 0.429 *** (8.11) |
| Distan | −0.069 ** (−2.59) | −0.109 *** (−3.89) |
| QoRoad |  | 0.153 *** (3.59) |
| QoPort |  | −0.116 ** (−2.63) |
| QoAir |  | 0.149 ** (2.95) |
| QoElec |  | −0.126 ** (−2.79) |
| PoS |  | 0.042 (1.07) |
| GovE |  | 0.128 (1.15) |
| RegQ |  | −0.555 *** (−7.25) |

**Table 4.** *Cont.*

|  | Model 1 (VTF) | Model 5 (VTF) |
|---|---|---|
| RulL |  | −0.122 (−0.98) |
| ContrC |  | 0.262 ** (2.97) |
| Owner |  | 0.074 * (2.18) |
| Equity |  | 0.077 * (2.05) |
| Loan |  | 0.082 * (2.08) |
| Year FE | YES | YES |
| N | 1023 | 1023 |
| R Square | 0.456 | 0.533 |

Notes: *** $p < 0.01$, ** $p < 0.05$, * $p < 0.1$; Year FE = Year fixed effect; N represents the number of observations.

By comparing Table 4 with Table 3, it can be seen that the coefficient and significance of the explanatory variable did not change significantly after the explained variable was replaced. Therefore, the core conclusion of this paper has good robustness.

## 5. Conclusions

### 5.1. Findings

International infrastructure investment plays an essential role in sustainable social development. This paper used panel data of 134 countries from 2009 to 2017 to empirically test the location choice and the influencing factors of international infrastructure investment. The conclusions of this study are as follows. First, international infrastructure investment tends to select countries or regions with good condition of highway infrastructure. Second, international infrastructure investment tends to choose countries or regions with low development level of port and power infrastructure, and market space is the important factor attracting international infrastructure investment. Third, bilateral diplomatic visits play a significant role in promoting international infrastructure investment. Fourth, international infrastructure investment significantly tends to choose resource-rich countries or regions, and resource exchange is the prominent motivation of international infrastructure investment.

The above results have significant policy implications. First, developing countries could take highway infrastructure promotion as the priority of economic development to attract foreign investment. Second, developing countries with poor port and power infrastructure conditions can take advantage of international technologies and capitals to develop port and power industries. Third, as senior leaders' diplomatic visits could help create investment opportunities and facilitate contracting in infrastructure project delivery, developing countries should establish dialogue platform to enhance cooperation between countries at diplomatic level. Fourth, developing countries should optimally transform their resource into their industrial competence in the process of collaborating with international infrastructure investors.

The study identified six major factors affecting international infrastructure in-vestment, and accordingly, established a set of models on the factors influencing choice of international infrastructure investment location, which revealed the influences of infrastructure quality, bilateral diplomatic visits, and resource endowment on China's investment in the engineering projects (CIIEP). The outcomes of the study not only provided a theoretical framework for engineering enterprises to make decisions on international infrastructure investment, but also to have policy implications for developing countries to attract international infrastructure investment.

### 5.2. Limitations and Future Research Directions

The factors influencing international infrastructure investment choices identified in this study could be further expanded. For instance, this study did not take into account informal institutional factors such as cultural, language, and religious differences. International infrastructure projects have long construction durations, large investment scales, and complex technology, and the interactions between different factors should be studied.

This study only conducted research on China's international investment in the infrastructure sector, and the data from more countries need to be collected to further verify the insights of this study. This study analyzed heterogeneity at the country level only, and it did not consider the effect of project-level or firm-level heterogeneity. Different projects and different firms can also be studied to further understand the features of international investment on infrastructure projects.

**Author Contributions:** Conceptualization, S.H., Y.W. and W.T.; methodology, S.H. and Y.W.; validation, Y.W.; formal analysis, S.H.; investigation, S.H., Y.W. and W.T.; data curation, W.T.; writing—original draft preparation, S.H.; writing—review and editing, Y.W. and W.T.; project administration, W.T. All authors have read and agreed to the published version of the manuscript.

**Funding:** This research was funded by the National Natural Science Foundation of China, grant numbers 72171128 and 51579135, and the State Key Laboratory of Hydroscience and Engineering, grant numbers 2022-KY-04.

**Institutional Review Board Statement:** Not applicable.

**Informed Consent Statement:** Not applicable.

**Data Availability Statement:** The data presented in this study are all available on request from the corresponding author.

**Conflicts of Interest:** The authors declare no conflict of interest.

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
