# Peer review of "Factors Influencing International Infrastructure Investment: An Empirical Study from Chinese Investors"

_sustainability, doi:10.3390/su151411072_

Round 1
Reviewer 1 Report
Please check attachment.

The article must be thoroughly proof-read by a native speaker to further improve clarity.
Author Response
Thank you very much for your carefully reviewing! The article has been thoroughly proof-read by a native speaker to further improve clarity. All your comments have been incorporated into our revised manuscript. Our responses are shown in the attachment.

Reviewer 2 Report
This paper constructs a panel data regression empirical model, conducts empirical research on issues related to international sustainable development infrastructure investment, and draws some innovative conclusions, but there are still the following deficiencies:
1. The introduction is short. The author needs to expand the background of the topic and explain the contribution in more detail.
2. The author needs to explain in the introduction how this study is different and novel compared to the results of previous studies.
3. The literature review in this paper is carried out from six aspects. Is it necessary to have a concluding literature review paragraph to clarify the previous research?
4. In the empirical design phase, what is the basis for selecting indicators? Can you cite relevant literature to prove that the selection of this indicator is justified?
5. When selecting control variables, literature discussing the relationship between control variables and explanatory variables can be included.
6. When reporting regression results, discussion with previous literature, such as in-depth research or challenges to research, can be included.
7. Research results could be discussed more fully.
8. This research conclusion can be more consistent with relevant policy explanations.
Author Response
Dear reviewer,
Thank you very much for your carefully reviewing! All your comments have been incorporated into our revised manuscript. Our responses are shown in the attachment.

Round 2
Reviewer 1 Report
The title and the overall article is in much better shape.
Improved